# Moderate-intensity physical activity reduces the role of serum PFAS on COPD: A cross-sectional analysis with NHANES data

Manyi Pan[1], Yuxin Zou[1], Gang Wei[2], Caoxu Zhang[3], Kai Zhang[4]*, Huaqi Guo◉[1]*, Weining Xiong◉[1]*

1 Department of Respiratory and Critical Care Medicine, Shanghai Ninth People's Hospital, Shanghai Jiao Tong University School of Medicine, Shanghai, China, 2 Beijing Key Laboratory of Diabetes Research and Care, Department of Endocrinology, Beijing Diabetes Institute, Beijing Tongren Hospital, Capital Medical University, Beijing, China, 3 Department of Molecular Diagnostics & Endocrinology, The Core Laboratory in Medical Center of Clinical Research, Shanghai Ninth People's Hospital, State Key Laboratory of Medical Genomics, Shanghai Jiao Tong University School of Medicine, Shanghai, China, 4 Department of Public health, Shanghai Jiao Tong University School of Medicine, Shanghai, China

* zhangk19@sjtu.edu.cn (KZ); Guohuaqi814102@163.com (HG); xiongdoctor@qq.com (WX)

**Data Availability Statement:** All files are available from the NHANES database.

## Abstract

### Background

Chronic obstructive pulmonary disease (COPD) has emerged as a leading cause of chronic disease morbidity and mortality globally, posing a substantial public health challenge. Per-fluoroalkyl substances (PFAS) are synthetic chemicals known for their high stability and durability. Research has examined their potential link to decreased lung function. Physical activity (PA) has been identified as one of the primary modalities of the non-pharmacological treatment of COPD.

### Methods

To investigate the relationship between PFAS and COPD, and whether physical activity could reduce the risk of COPD caused by PFAS exposure, we used data from the NHANES 2013–2018, a cross-sectional study. Logistic regression analysis was used to examine the associations between PFAS and COPD in adult populations, and their associations in different PA types.

### Results

We finally included 4857 participants in the analysis, and found that Sm-PFOS (OR: 1.250), PFOA (OR: 1.398) and n-PFOA (OR: 1.354) were closely related to COPD; After stratified by gender, age and smoking, the results showed that Sm-PFOA (OR: 1.312) was related to COPD in female adult, and PFOA (OR: 1.398) and n-PFOA (OR: 1.354) were associated with COPD in male adults; The associations of Sm-PFOS (OR: 1.280), PFOA (OR: 1.481) and n-PFOA (OR: 1.424)with COPD tended to be stronger and more consistent in over 50 years old adults; Sm-PFOS was related to COPD in current smoker (OR: 1.408), and PFOA was related to COPD in former smoker (OR: 1.487); Besides, in moderate-intensity PA

**Funding:** This work was financially supported by the National Natural Science Foundation of China (No. 82090015).

**Competing interests:** The authors declare that there is no conflict of interest regarding the publication of this article.

group, there were no associations of Sm-PFOS, PFOA and n-PFOA with COPD stratified by gender, age and smoking.

## Conclusion

PFAS exposure may increase the risk of developing COPD, but regular moderate-intensity physical activity can protect individuals from evolving to the disease. However, longitudinal studies are needed to support these preliminary findings.

## 1. Introduction

Chronic obstructive pulmonary disease (COPD) is a preventable and treatable respiratory disease characterized by persistent symptoms and airflow limitation [1, 2]. It has become a significant cause of chronic morbidity and mortality worldwide, and represents a major public health challenge [3, 4]. The global burden of COPD projected to increase in the coming decades due to continued exposure to COPD risk factors [3]. Perfluoroalkyl substances (PFAS), synthetic chemicals known for their high stability and durability, have been widely used in food contact paper, carpets, furniture, and other household products [5–7]. The National Health and Nutrition Examination Survey (NHANES, 2013–2014) has documented measurable serum levels of perfluorooctane sulfonate (PFOS), perfluorooctanoate (PFOA), perfluorohexane sulfonic acid (PFHxS), and perfluorononanoic acid (PFNA) in over 95% of U.S. general population [8, 9]. Available studies have shown that PFAS may be related to reduced lung function in childhood and adolescence [4, 10, 11]. Pulmonary function decreases to varying degrees with age, especially in older stages [12], and COPD, a typical lung function reduction-related disease, is characterized by non-full irreversible airflow limitation (FEV1/FVC<0.7 post-bronchodilation) [13, 14], and the population suffering from COPD is mostly elderly [15, 16]. PFAS are now have been considered one class of persistent pollutants due to their apparent bioaccumulation and long-range transport potential [17]. Therefore, we hypothesized circumstantially that PFAS levels in blood would be essential for assessing their association with COPD, and we were the first to investigate the relationship between PFAS and COPD.

Physical activity (PA) includes any bodily movement produced by skeletal muscles that requires energy expenditure. It encompasses various activities across four domains: occupational (work-related), domestic (household chores), transportation (walking or cycling to get from one place to another), and leisure time (recreational activities) [18, 19]. PA is recognized as a primary non-pharmacological treatment for COPD and can effectively improve exercise tolerance, clinical symptoms and quality of life of COPD patients [19]. Meanwhile, epidemiological studies by population cohorts have reported longitudinal associations between PA and lung function, attenuated lung function decline and lower COPD incidence [20–22].

Since PFAS and PA can individually affect COPD in opposite directions, it remains unclear whether they have a combined role in regulating COPD. Therefore, we performed an analysis of 2013–2018 National Health and Nutrition Examination Survey (NHANES) data to investigate the association between PFAS and COPD in U.S. adults. Additionally, we assessed whether PA can mitigate the risk of COPD due to PFAS exposure and evaluated the effects of different PA intensities. This study aims to elucidate the relationship between PFAS and COPD and provide scientific evidence for the prevention of COPD through PA in the presence of PFAS exposure.

## 2. Materials and methods

### 2.1. Study design and population

The NHANES, administered by the National Center for Health Statistics (NCHS) of the Centers for Disease Control and Prevention (CDC), is a cross-sectional study employing a multistage, random probability sampling design. It was approved by the National Center for Health Statistics (NCHS) Institutional Review Board, and informed consent was obtained from all participants. Health interviews were completed by participants in the household, and physical examinations were conducted at the equipped mobile examination center, where urine and blood samples were collected.

Three sections (2013–2014, 2015–2016 and 2017–2018) with information on demographics, serum PFAS concentration, PA, and COPD (https://www.cdc.gov/nchs/nhanes/index.htm) were combined and analyzed. The detailed process of the participants included in this study is shown in Fig 1. Ultimately, we included 4857 participants aged 18 years and older in the analysis.

### 2.2. Measurement of PFAS

Nine types of PFAS were detected in the NHANES 2013–2014, 2015–2016, and 2017–2018 cycles, including linear perfluorooctane sulfonate (n-PFOS); monomethyl branched isomers of PFOS (Sm-PFOS); linear perfluorooctanoate (n-PFOA); branched isomers of perfluorooctanoate (Sb-PFOA); pefluorodecanoic acid (PFDA); perfluorohexane sulfonic acid (PFHxS); 2- (N-Methyl- perfluorooctane sulfonamide) acetic acid (nm-PFOSA); perfluorononanoic acid (PFNA) and perflurododecanoic acid (PFDoA). The measurement of PFAS in serum was performed using online-solid phase extraction-high performance liquid chromatography-turbo ion spray ionization-tandem mass spectrometry. Details regarding the analysis methodology are available on the NHANES website (https://wwwn.cdc.gov/Nchs/Nhanes/2017-2018/PFAS_J.htm).

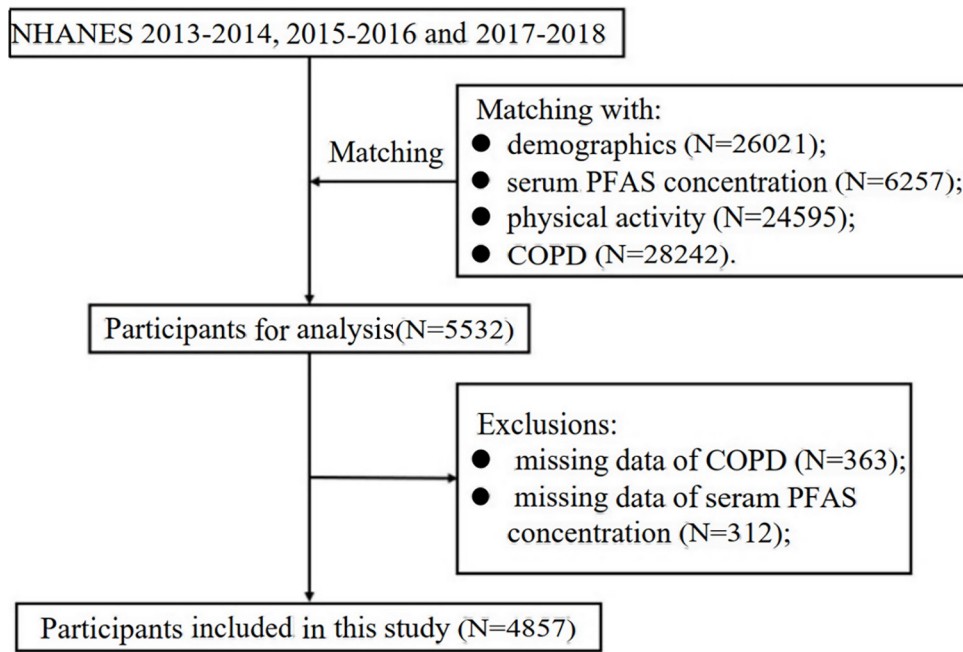

**Fig 1. Three sections (2013–2014, 2015–2016 and 2017–2018) with information on demographics.**

In addition, we calculated total PFOA (PFOA) as the sum of n-PFOA and Sb-PFOA, and total PFOS (PFOS) as the sum of n-PFOS and Sm-PFOS. Following NHANES analysis guidelines, the values below the LLOD were instead of $LLOD/\sqrt{2}$.

## 2.3. MET and PA calculation

The metabolic equivalent (MET) represents the oxygen consumption necessary to sustain resting metabolism. Based on the energy consumption in a quiet and sitting position, it is a commonly used index to express the relative energy metabolism level. Physical activity (PA) data were collected using the NHANES Physical Activity Questionnaire (PAQ), which assesses participants' activity levels by inquiring about the frequency, duration, and intensity of various activities performed over a defined period, typically the past week or month. Activities covered include walking, jogging, sports, and household chores, with participants providing details on the days, hours, and minutes spent in each activity. Trained interviewers completed the PAQ at home using the Computer Assisted Personal Interview (CAPI) system. The MET values are varied among different types of physical activities, and NHANES provides recommended MET values for each type. The PAQ survey included Vigorous work-related activity (MET = 8), Moderate work-related activity (MET = 4), Walking or bicycling for Transportation (MET = 4), Vigorous leisure-time PA (MET = 8) and Moderate leisure-time PA (MET = 4). PA can be calculated based on the MET value, activity type, weekly frequency, and duration. We computed the value of PA from the following formula: PA (MET-min/week) = MET × weekly frequency × duration of each physical activity [23, 24]. Subsequently, participants were divided into three groups based on the standard scoring criteria of the International Physical Activity Questionnaire (IPAQ): low (<600 MET-min/week), moderate (600 to 3000 MET-min/week), and high (≥3000 MET-min/week) [23]. The threshold of 600 MET-min/week corresponds to the WHO recommended guidelines for moderate-intensity PA (150 min per week) [25–27].

## 2.4. Outcome and covariables

COPD diagnosis was determined using the medical conditions data file (MCQ_H) with the question: "Has a doctor or other health professional ever told you that you/she/he had COPD?". Those who answered "yes" were categorized as COPD, and who answered "no" were categorized as non-COPD.

We adjusted for variables associated with COPD diagnosis or believed to confound the relationship between serum PFAS concentrations and COPD diagnosis. Age (continuous), Race (categorical), Gender (categorical), BMI (categorical), PA (categorical), Covered by health insurance (categorical), NHANES cycle (categorical), Drinking (categorical), Marital (categorical), Household income (categorical) [28, 29], were extracted from the demographic variables and sample weights data file (DEMO_H).

## 2.5. Statistical analysis

All analyses were performed using R software. PFOA and PFOS were analyzed as the sum of the respective linear and branched polymers (Total PFOS (PFOS) = Sm-PFOS + n-PFOS, Total PFOA (PFOA) = Sb-PFOA + n-PFOA). Continuous variables were represented by Median ± Inter quartile range (Median ± IQR), and categorical variables were denoted by N (%). Differences in continuous variables between COPD and non-COPD participants were assessed using t-tests. Moreover, chi-square tests were applied to evaluate the differences in categorical demographic. Serum PFAS distributions were usually right-skewed. Therefore, ln-

transformation was performed in the regression analysis to ensure the linearity of the relationship between the predictors and the logit of the outcome. The associations between PFAS and COPD risk were investigated by univariate and multivariate logistic regression analysis. The results were presented with odds ratios (OR) and 95% confidence intervals (95% CI). First, the study of the Crude model was carried out. Then, Model 1 adjusted for age, race, gender, BMI, smoking, PA, covered by health insurance, cycle (referring to NHANES Cycle). Model 2 adjusted for drinking, marital, household income. Then, we analyzed the relationships between serum PFAS concentrations and COPD risk among subpopulation stratified by age, gender or smoking, adjusting for age, race, gender, BMI, smoking, physical activity, covered by health insurance, Cycle, drinking, marital, household income. Moreover, to explore the effect of various types of PA(Low-intensity physical activity, Moderate-intensity physical activity, High-intensity physical activity) on the relationship between serum PFAS concentrations and COPD risk, we screened serum PFAS concentrations that are correlated with COPD among subpopulation stratified by age, gender or smoking, and used multivariate logistic regression to analyze the correlation between serum PFOS concentration and COPD at different intensities of physical activity. The probabilities (P values) were bilateral for all reports, and P<0.05 was regarded as statistically significant.

## 3. Results

### 3.1. Basic characteristics of participants

The characteristics of the study participants are summarized in Table 1. The final number of participants included in the analysis was 4857, comprising 2304 men and 2553 women. With all participants being adults ≥18 years old, 1962 (40.4%) were between the ages of 18 and 44, 1713 (35.27%) were between the ages of 45 and 64, and the rest were over 65 years old. There were 929 (19.15%) current smokers and 1120 (23.08%) former smokers. In addition, 1967 (40.5%) made low-intensity PA, 1363 (28.1%) had moderate-intensity PA, 1495 (30.8%) had high-intensity PA, and 200 was diagnosed with COPD.

Among all participants, the distributions of several key characteristics were markedly higher in COPD patients compared to non-COPD patients. These characteristics included age (65 and over), race (Non-Hispanic White), household income (<$20,000), marital status (Divorced and Widowed), smoking status (Current smoker and Former smoker), physical activity level (Low-intensity), health insurance coverage (Yes), and type of health insurance (Medicare).

### 3.2. Serum PFAS and COPD

The associations between individual PFAS and COPD in adults are presented in Table 2. After adjusting confounding variables (Age, Race, Gender, BMI, Smoking, PA, covered by health insurance and Cycle) in Model 1, an ln-unit increase in Sm-PFOS was associated with COPD (OR: 1.189, 95% CI: 1.008–1.409). In Model 2, we further adjusted for Drinking, Marital and Household income, and found that the relationship between Sm-PFOS and COPD persisted (an ln-unit increase, OR: 1.250, 95% CI: 1.040–1.511). Additionally, both PFOA (an ln-unit increase, OR: 1.398, 95% CI: 1.078–1.819) and n-PFOA (an ln-unit increase, OR: 1.354, 95% CI: 1.068–1.729) were shown to be closely related to COPD. Thus, the concentrations of serum Sm-PFOS, PFOA and n-PFOA are associated with COPD, respectively.

### 3.3. Serum PFAS and COPD stratified by gender, age and smoking

Stratification by gender showed that an ln-unit increase in Sm-PFOS was associated with COPD in female adults (OR: 1.312, 95% CI: 1.015–1.721), but not in male adults; For PFOA

**Table 1. Basic characteristics of participants.**

| Variables | Total | Non-COPD | COPD | P |
|---|---|---|---|---|
| | (n = 4857) | (n = 4657) | (n = 200) | |
| Gender, n (%) | | | | 0.124 |
| Men | 2304 (47.44) | 2198 (47.20) | 106 (53.00) | |
| Women | 2553 (52.56) | 2459 (52.80) | 94 (47.00) | |
| Age.Group, n (%) | | | | < 0.001 |
| 18–44 | 1962 (40.40) | 1953 (41.94) | 9 (4.50) | |
| 45–64 | 1713 (35.27) | 1629 (34.98) | 84 (42.00) | |
| 65 and over | 1182 (24.34) | 1075 (23.08) | 107 (53.50) | |
| Race, n (%) | | | | < 0.001 |
| Mexican American | 723 (14.89) | 715 (15.35) | 8 (4.00) | |
| Non-Hispanic Black | 1048 (21.58) | 1017 (21.84) | 31 (15.50) | |
| Non-Hispanic White | 1797 (37.00) | 1673 (35.92) | 124 (62.00) | |
| Other Hispanic | 519 (10.69) | 504 (10.82) | 15 (7.50) | |
| Other Race—Including Multi-Racial | 770 (15.85) | 748 (16.06) | 22 (11.00) | |
| Household_income, n (%) | | | | < 0.001 |
| <20,000$ | 917 (20.19) | 842 (19.32) | 75 (40.98) | |
| ≥75,000$ | 1273 (28.03) | 1261 (28.93) | 12 (6.56) | |
| 20,000–44,999$ | 1494 (32.89) | 1428 (32.76) | 66 (36.07) | |
| 45,000–74,999$ | 858 (18.89) | 828 (19.00) | 30 (16.39) | |
| Marital_status, n (%) | | | | < 0.001 |
| Divorced | 529 (10.90) | 489 (10.50) | 40 (20.00) | |
| Living with partner | 396 (8.16) | 392 (8.42) | 4 (2.00) | |
| Married | 2491 (51.31) | 2404 (51.64) | 87 (43.50) | |
| Never married | 886 (18.25) | 856 (18.39) | 30 (15.00) | |
| Separated | 180 (3.71) | 174 (3.74) | 6 (3.00) | |
| Widowed | 373 (7.68) | 340 (7.30) | 33 (16.50) | |
| BMI, n (%) | | | | 0.782 |
| Under 25 kg/m$^2$ | 1344 (28.00) | 1293 (28.00) | 51 (26.30) | |
| 25–30.0 kg/m$^2$ | 1534 (31.90) | 1468 (31.80) | 66 (34.00) | |
| 30.0 kg/m$^2$ and over | 1928 (40.10) | 1851 (40.10) | 77 (39.70) | |
| Smoking, n (%) | | | | < 0.001 |
| Current smoker | 929 (19.15) | 842 (18.10) | 87 (43.50) | |
| Former smoker | 1120 (23.08) | 1037 (22.29) | 83 (41.50) | |
| Never smoker | 2803 (57.77) | 2773 (59.61) | 30 (15.00) | |
| Drinking, n (%) | | | | 0.033 |
| Drinker | 1576 (36.30) | 1523 (36.65) | 53 (28.65) | |
| Non-drinker | 2765 (63.70) | 2633 (63.35) | 132 (71.35) | |
| Physical activity (PA), n (%) | | | | < 0.001 |
| Low-intensity physical activity (PA = 0) | 1967 (40.50) | 1853 (39.80) | 114 (57.00) | |
| Moderate-intensity physical activity (PA = 1) | 1363 (28.10) | 1314 (28.20) | 49 (24.50) | |
| High-intensity physical activity (PA = 2) | 1495 (30.80) | 1460 (31.40) | 35 (17.50) | |
| Covered_by_health_insurance, n (%) | | | | < 0.001 |
| No | 857 (17.67) | 847 (18.21) | 10 (5.03) | |
| Yes | 3993 (82.33) | 3804 (81.79) | 189 (94.97) | |
| Type_of_health_insurance, n (%) | | | | < 0.001 |
| Medicaid | 646 (14.10) | 597 (13.59) | 49 (26.06) | |
| Medicare | 883 (19.28) | 796 (18.12) | 87 (46.28) | |

*(Continued)*

**Table 1.** (Continued)

| Variables | Total | Non-COPD | COPD | P |
|---|---|---|---|---|
| | (n = 4857) | (n = 4657) | (n = 200) | |
| None | 857 (18.71) | 847 (19.29) | 10 (5.32) | |
| Other government | 188 (4.10) | 178 (4.05) | 10 (5.32) | |
| Private | 1653 (36.09) | 1634 (37.20) | 19 (10.11) | |
| Single service plan | 353 (7.71) | 340 (7.74) | 13 (6.91) | |
| Education_level_all, n (%) | | | | < 0.001 |
| <High school | 1081 (22.29) | 1026 (22.07) | 55 (27.50) | |
| College graduate or above | 1160 (23.92) | 1142 (24.56) | 18 (9.00) | |
| High school graduate | 1082 (22.31) | 1018 (21.90) | 64 (32.00) | |
| Some college | 1526 (31.47) | 1463 (31.47) | 63 (31.50) | |

and n-PFOA, the positive effect estimates were found in male adults (PFOA, an ln-unit increase, OR: 1.398, 95% CI: 1.078–1.819; n-PFOA, an ln-unit increase, OR: 1.354, 95% CI: 1.068–1.729), but not in female adult (Table 3). Then, stratification by age (50 years old as the limit) indicated stronger and more consistent positive effect estimates for Sm-PFOS, PFOA, and n-PFOA in over 50 years old adults (Sm-PFOS, an ln-unit increase, OR: 1.280, 95% CI: 1.052–1.569; PFOA, an ln-unit increase, OR: 1.481, 95% CI: 1.125–1.960; n-PFOA, an ln-unit increase, OR: 1.424, 95% CI: 1.107–1.847), but not in less than 50 years old adult (Table 4). Finally, stratification by smoking status (including never smoker, former smoker and current smoker) indicated that an ln-unit increase in Sm-PFOS was correlated with COPD in current smoker (OR: 1.408, 95% CI: 1.016–1.992), and an ln-unit increase in PFOA was associated with COPD in former smoker (OR: 1.487, 95% CI: 1.003–2.219) (Table 5).

**Table 2.** Associations between serum PFAS and COPD [Odds ratios (95% confidence intervals)].

| | Model 1 | Model 2 |
|---|---|---|
| PFOS | 0.998(0.862,1.159) | 1.021(0.867,1.205) |
| n-PFOS | 0.998(0.862,1.159) | 1.021(0.867,1.205) |
| Sm-PFOS | 1.189(1.008,1.409) * | 1.250(1.040,1.511) * |
| PFOA | 1.242(0.987,1.567) | 1.398(1.078,1.819) ** |
| n-PFOA | 1.214(0.986,1.502) | 1.354(1.068,1.729) ** |
| Sb-PFOA | 0.585(0.171,1.795) | 0.553(0.147,1.877) |
| PFDA | 0.842(0.676,1.037) | 0.862(0.681,1.077) |
| PFHxS | 1.015(0.859,1.202) | 1.001(0.835,1.204) |
| nm-PFOSA | 1.050(0.821,1.320) | 1.049(0.804,1.343) |
| PFNA | 1.048(0.869,1.267) | 1.086(0.884,1.337) |
| PFDoA | 3.003(0.700,7.528) | 2.233(0.381,9.702) |

Note: PFAS was ln-transformed

Model 1 adjusted for Age (continuous), Race (categorical), Gender (categorical), BMI (categorical), Smoking (categorical), PA (categorical), Covered by health insurance (categorical), Cycle (categorical)

Model 2 adjusted for Drinking (categorical), Marital (categorical), Household income (categorical).

*P<0.05

**P<0.01.

**Table 3. Associations between serum PFAS and COPD in female and male adults [Odds ratios (95% confidence intervals)].**

|  | Female | Male |
|---|---|---|
| PFOS | 1.144(0.903,1.458) | 0.934(0.737,1.191) |
| n-PFOS | 1.144(0.903,1.458) | 0.934(0.737,1.191) |
| Sm-PFOS | 1.312(1.015,1.721) * | 1.219(0.926,1.622) |
| PFOA | 1.173(0.808,1.710) | 1.657(1.135,2.455) ** |
| n-PFOA | 1.165(0.830,1.651) | 1.563(1.105,2.253) ** |
| Sb-PFOA | 1.260(0.166,6.062) | 2.842(0.05,1.484) |
| PFDA | 0.787(0.556,1.086) | 0.905(0.639,1.256) |
| PFHxS | 0.943(0.725,1.235) | 1.048(0.810,1.363) |
| nm-PFOSA | 1.086(0.746,1.532) | 0.989(0.656,1.430) |
| PFNA | 1.091(0.803,1.492) | 1.093(0.821,1.460) |
| PFDoA | 0.705(0.375,1.208) | 1.050(0.648,1.619) |

Note: PFAS was ln-transformed

Adjusted for Age (continuous), Race (categorical), BMI (categorical), Smoking (categorical), PA (categorical),

Covered by health insurance (categorical), Cycle (categorical), Drinking (categorical), Marital (categorical),

Household income (categorical).

*P<0.05

**P<0.01.

## 3.4. Regulation of PA on PFAS and COPD

Based on the relationship between PFAS and COPD stratified by gender, age, and smoking, we further explored these associations in different PA groups (Table 6). we further explored these associations in different PA groups (Table 6). We found that male adults in the low-intensity PA group (PA = 0) were at higher risk of COPD associated with PFOA (OR: 2.312, 95% CI:

**Table 4. Associations between serum PFAS and COPD in < 50 and ≥ 50 year adult [Odds ratios (95% confidence intervals)].**

|  | < 50 | ≥ 50 |
|---|---|---|
| PFOS | 1.308(0.635,2.827) | 1.026(0.866,1.219) |
| n-PFOS | 1.308(0.635,2.827) | 1.026(0.866,1.219) |
| Sm-PFOS | 1.503(0.809,3.063) | 1.280(1.052,1.569)* |
| PFOA | 1.363(0.597,3.290) | 1.481(1.125,1.960)** |
| n-PFOA | 1.318(0.650,2.911) | 1.424(1.107,1.847)** |
| Sb-PFOA | 1.823(0.006,51.595) | 0.427(0.103,1.580) |
| PFDA | 0.963(0.363,2.187) | 0.871(0.680,1.102) |
| PFHxS | 0.973(0.500,2.005) | 1.028(0.849,1.248) |
| nm-PFOSA | 0.439(0.049,1.734) | 1.153(0.878,1.490) |
| PFNA | 0.856(0.430,1.742) | 1.184(0.954,1.473) |
| PFDoA | 1.298(0.337,3.875) | 0.851(0.576,1.208) |

Note: PFAS was ln-transformed

Adjusted for Race (categorical), Gender (categorical), BMI (categorical), Smoking (categorical), PA (categorical),

Covered by health insurance (categorical), Cycle (categorical), Drinking (categorical), Marital (categorical),

Household income (categorical).

*P<0.05

**P<0.01.

**Table 5. Associations between serum PFAS and COPD in adults stratified by smoking status [Odds ratios (95% confidence intervals)].**

| | Never smoker | Former smoker | Current smoker |
|---|---|---|---|
| PFOS | 0.845(0.563,1.307) | 0.948(0.731,1.234) | 1.113(0.858,1.461) |
| n-PFOS | 0.845(0.563,1.307) | 0.948(0.731,1.234) | 1.113(0.858,1.461) |
| Sm-PFOS | 1.398(0.883,2.338) | 1.128(0.863,1.492) | 1.408(1.016,1.992)* |
| PFOA | 1.595(0.846,2.995) | 1.487(1.003,2.219)* | 1.429(0.919,2.267) |
| n-PFOA | 1.538(0.857,2.805) | 1.427(0.996,2.073) | 1.392(0.932,2.131) |
| Sb-PFOA | 1.570(0.067,8.723) | 2.355(0.034,1.325) | 0.693(0.046,9.846) |
| PFDA | 0.634(0.316,1.127) | 0.924(0.659,1.270) | 0.863(0.559,1.291) |
| PFHxS | 1.168(0.765,1.823) | 1.183(0.894,1.585) | 9.755(0.549,1.035) |
| nm-PFOSA | 0.546(0.188,1.211) | 1.147(0.777,1.629) | 1.146(0.732,1.739) |
| PFNA | 1.071(0.667,1.744) | 1.068(0.776,1.473) | 1.180(0.823,1.711) |
| PFDoA | 1.123(0.512,2.079) | 0.783(0.433,1.302) | 0.834(0.425,1.514) |

Note: PFAS was ln-transformed

Adjusted for Age (continuous), Race (categorical), Gender (categorical), BMI (categorical), PA (categorical), Covered by health insurance (categorical), Cycle (categorical), Drinking (categorical), Marital (categorical), Household income (categorical).

*P<0.05.

1.051, 5.665), female adults in the low-intensity PA group (PA = 0) were at higher risk of COPD associated with Sm-PFOS (OR: 1.616, 95% CI: 1.012, 2.73), and male adults in high-intensity PA group (PA = 2) were at highest risk of COPD associated with PFOA (OR: 3.111, 95% CI: 1.217, 8.872) and n-PFOA (OR: 2.92, 95% CI: 1.206, 7.912); For adults over the age of 50, the risks of COPD associated with Sm-PFOS(OR: 1.448, 95% CI: 1.031, 2.078), PFOA(OR: 2.103, 95% CI: 1.051, 5.665) and n-PFOA(OR: 1.935, 95% CI: 1.231, 3.158) were highest in the low-intensity PA group (PA = 0); For current smoker, high-intensity PA group (PA = 2) were at the highest risk of COPD associated with Sm-PFOS(OR: 4.862, 95% CI: 1.664, 25.292), and For former smoker, high-intensity PA group (PA = 2) were at the highest risk of COPD associated with PFOA (OR: 7.661, 95% CI: 2.147, 34.929); Additionally, in the moderate-intensity PA group (PA = 1), there were no associations between Sm-PFOS, PFOA, and n-PFOA and COPD in adult populations stratified by gender, age, and smoking, indicating that moderate-intensity PA could mitigate PFAS-related COPD.

## 4. Discussion

In this study indicated that PFAS (including Sm-PFOS, PFOA and n-PFOA) were associated with COPD in adult populations. Interestingly, these associations varied across different adult populations stratified by gender, age and smoking status. For example, Sm-PFOA was related to COPD in female adults, and PFOA and n-PFOA were associated with COPD in male adults; Sm-PFOS, PFOA and n-PFOA were more strongly and consistently associated with COPD in over 50 years old adults, but not in less than 50 years old adults; Sm-PFOS was related to COPD in current smoker, and PFOA was associated with COPD in former smoker. Furthermore, moderate-intensity PA appeared to mitigate the associations of Sm-PFOS, PFOA, and n-PFOA with COPD in adult populations.

Currently, there are relatively few studies exploring the associations between PFAS and respiratory diseases, mainly focused on asthma, airway inflammation and infection [30–32]. This study is the first to reveal the consistent and positive correlation between PFAS (including Sm-PFOS, PFOA, and n-PFOA) and COPD in adult populations. Actually, the presence of incomplete reversible airflow limitation is required for COPD to be diagnosed by performing

**Table 6. Associations between serum PFAS and COPD in different PA.**

|  | PA | OR (95% CI) |
|---|---|---|
| **Sm-PFOS** |  |  |
| Gender, Female | PA = 0 | 1.616 (1.012, 2.73)* |
| Gender, Female | PA = 1 | 1.281 (0.864, 1.961) |
| Gender, Female | PA = 2 | 2.433 (0.965, 8.475) |
| Age, ≥50 | PA = 0 | 1.448 (1.031, 2.078)* |
| Age, ≥50 | PA = 1 | 1.153 (0.853, 1.587) |
| Age, ≥50 | PA = 2 | 1.453 (0.9, 2.431) |
| Smoking, Current smoker | PA = 0 | 1.977 (0.689, 7.387) |
| Smoking, Current smoker | PA = 1 | 1.171 (0.725, 1.942) |
| Smoking, Current smoker | PA = 2 | 4.862 (1.664, 25.292)* |
| **PFOA** |  |  |
| Gender, Male | PA = 0 | 2.312 (1.051, 5.665)* |
| Gender, Male | PA = 1 | 1.171 (0.657, 2.112) |
| Gender, Male | PA = 2 | 3.111 (1.217, 8.872)* |
| Age, ≥50 | PA = 0 | 2.103 (1.28, 3.569)* |
| Age, ≥50 | PA = 1 | 1.163 (0.763, 1.773) |
| Age, ≥50 | PA = 2 | 1.854 (0.861, 4.186) |
| Smoking, former smoker | PA = 0 | 1.521 (0.671, 3.737) |
| Smoking, former smoker | PA = 1 | 0.912 (0.5, 1.652) |
| Smoking, former smoker | PA = 2 | 7.661 (2.147, 34.929)* |
| **n-PFOA** |  |  |
| Gender, Male | PA = 0 | 1.994 (0.989, 4.538) |
| Gender, Male | PA = 1 | 1.155 (0.686, 1.986) |
| Gender, Male | PA = 2 | 2.92 (1.206, 7.912)* |
| Age, ≥50 | PA = 0 | 1.935 (1.231, 3.158)* |
| Age, ≥50 | PA = 1 | 1.146 (0.782, 1.694) |
| Age, ≥50 | PA = 2 | 1.795 (0.877, 3.887) |

Note: PFAS was ln-transformed

Adjusted for Age (continuous), Race (categorical), Gender (categorical), BMI (categorical), PA (categorical), Covered by health insurance (categorical), Cycle (categorical), Drinking (categorical), Marital (categorical), Household income (categorical).

*P<0.05.

spirometry and a post-bronchodilator forced expiratory volume in one second (FEV1)/forced vital capacity (FVC) ≤0.7 [33]. It has been proved that the concentrations of PFOA and PFOS in cord blood were correlated with decreasing lung function in childhood [34], indirectly supporting the associations of Sm-PFOS, PFOA, and n-PFOA with COPD.

As adults age, their lung function tends to increase first and then decrease, and by age 50, the rate of decline slows down [35, 36]. Therefore, this study stratified COPD patients by age, and indicated that Sm-PFOS, PFOA and n-PFOA were related to COPD in over 50 years old adults. Still, these associations were not shown in less than 50 years old adults, which indicated that people over 50 years old should pay more attention to avoid the harm of PFAS related to COPD. Given that the most intensively studied risk factor for COPD was cigarette smoking [37], this study divided the COPD patients into three groups: non-smoker, former smoker and current smoker, then explored the associations of Sm-PFOS, PFOA and n-PFOA with these three COPD patients. And we found that Sm-PFOS was related to COPD in current smoker,

but not in non-smoker or former smoker, PFOA was related to COPD in the former smoker, but not in non-smoker or current smoker, indicating that smoking could promote the occurrence of PFAS-related COPD, and even smoking in the past and not smoking now has a promoting effect. Thus, to prevent the occurrence of PFAS-related COPD, smoking should be banned, because smoking could also lead to the event of PFAS-related COPD. In addition, gender-related differences in immune pathways and airway damage patterns [38, 39]; therefore, we further evaluated the associations of Sm-PFOS, PFOA and n-PFOA with COPD in female and male adults, respectively. And we found that PFOA and n-PFOA were associated with COPD in male adults, but not in female adults; and Sm-PFOS was associated with COPD in female adults, but not in male adults. These results indicated that male adults should pay more attention to avoiding the harm of COPD related to PFOA, and female adults should pay more attention to preventing the harm of COPD related to PFOS. As for the underlying mechanisms of the differently outcomes for Sm-PFOS, PFOA and n-PFOA in female and male adults, respectively, further scrupulously studies are needed.

The mechanism by which PFAS causes COPD has not been elucidated, but limited evidence suggests that the inflammatory response may be a critical link in mediating PFAS-induced COPD. On the one hand, the chronic inflammatory response could not only induce the destruction of parenchymal tissue leading to emphysema, but also disrupt the regular repair and defence mechanisms of the lung leading to small airway fibrosis, which eventually resulted in gas trapping and progressive airflow limitation, the pathological manifestations of COPD [40]; On the other hand, PFAS exposure could induce an inflammatory response in multiple organs. For example, a study using data from the NHANES 2005–2012 found that PFOA and PFOS were significantly associated with percentage increases in lymphocyte counts, and increased serum albumin, meaning these chemicals were related to chronic inflammation [41]; the odds of developing nonalcoholic steatohepatitis (NASH) increased significantly for each interquartile range (IQR) increase of PFOS compared to children with steatosis alone [42]; higher PFAS was associated with a lower degree of gut inflammation [43]; PFAS concentration was associated with a positive marker "FeNO>25 ppb" of eosinophilic airways inflammation [30].

Additionally, we found an interesting result that the low- or high-intensity PA group (PA = 0 or 2) was at the highest risk of COPD related to serum PFAS, and moderate-intensity PA has no risk of COPD related to serum PFAS. This finding is consistent with the study of David et al., which discovered that high-intensity PA did not produce any reduced risk for COPD [44]. This phenomenon may be caused by the fact that PA is a well-recognized determinant that reduces the incidence of COPD and inhibits the progression of COPD [29, 45], but high-intensity PA can bring burden to the body [46], which may have no effect on COPD or further exacerbate COPD. Therefore, our results were very plausible, indicating that moderate-intensity PA could eliminate the risk of COPD under PFAS exposure.

There were some limitations of this study. First, this study relied on self-reported questionnaires to identify COPD, which may result in missed diagnosis of the outcome, thus potentially attenuating these results. Second, although PA is a habitual human behavior and the data collected are reliable, recall bias cannot be ruled out for some of the data because they were collected through questionnaires. Additionally, for populations younger than 50 years, the number of people diagnosed with COPD will be smaller, which can lead to extremely imbalance between the COPD and non-COPD groups, resulting in biased and inaccurate results in the statistical analysis. To reduce this bias, we stratified by age (limited to 50 years), showing stronger positive effect estimates for PFAS in adults over 50 years old (Table 3), consistent with the results on the association between individual PFAS and COPD (Table 2). Finally, NHANES is a cross-sectional survey which limits the ability to establish the temporality of the exposure and outcome sequence. PFAS can persist in the body for years [47].

COPD is now among the top three causes of death worldwide, with 90% of these deaths occurring in low- and middle-income countries [48]. More than 3 million people died of COPD in 2012, accounting for 6% of all deaths globally [3]. COPD is a major cause of chronic morbidity and mortality throughout the world, representing a significant public health challenge [3]. And in our study, we found that the serum of the participants contained various types of PFAS contaminants. More importantly, PFAS has been classified as persistent organic pollutants (POPs) and can bind to proteins in the body, affecting homeostasis and ultimately leading to disease [47]. Thus, this association warrants further investigation. Future prospective cohort studies could explore the causal relationship between PFAS exposure and COPD development.

## 5. Conclusion

Sm-PFOS, PFOA and n-PFOA were associated with COPD in adults. PFOA and n-PFOA were associated with COPD in male adults, the associations of Sm-PFOS, PFOA and n-PFOA with COPD were stronger and more consistent in over 50 years old adults, and Sm-PFOS and PFOA were related to COPD in the current or former smoker. In addition, moderate-intensity PA could reduce the risk of COPD under PFAS exposure.

## Supporting information

**S1 Checklist. STROBE statement—Checklist of items that should be included in reports of *cross-sectional studies*.**
(DOC)

## Acknowledgments

We extend our appreciation to the editor and the anonymous reviewers for their invaluable and constructive comments. We would like to thank the National Health and Nutrition Examination Survey (NHANES) for providing the data used in this study.

## Author Contributions

**Conceptualization:** Yuxin Zou, Huaqi Guo.

**Data curation:** Huaqi Guo.

**Formal analysis:** Huaqi Guo.

**Funding acquisition:** Huaqi Guo, Weining Xiong.

**Investigation:** Huaqi Guo.

**Resources:** Caoxu Zhang.

**Software:** Gang Wei, Caoxu Zhang.

**Supervision:** Gang Wei, Caoxu Zhang, Kai Zhang, Weining Xiong.

**Validation:** Gang Wei, Caoxu Zhang, Kai Zhang.

**Visualization:** Gang Wei, Caoxu Zhang, Kai Zhang.

**Writing – original draft:** Manyi Pan, Yuxin Zou.

**Writing – review & editing:** Gang Wei, Kai Zhang, Weining Xiong.

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
