## [Decision Letter · Decision Letter 0]

19 Apr 2024

PONE-D-24-04889Moderate-intensity physical activity reduces the role of serum PFAS on COPD: A cross-sectional analysis with NHANES dataPLOS ONE

Dear Dr. Guo,

Thank you for submitting your manuscript to PLOS ONE. After careful consideration, we feel that it has merit but does not fully meet PLOS ONE’s publication criteria as it currently stands. Therefore, we invite you to submit a revised version of the manuscript that addresses the points raised during the review process.

The manuscript has been evaluated by two reviewers, and their comments are available below. Both reviewers indicate that work should be done to improve the reporting of this study. Specifically, clarification around the statistical methods used has been requested and revisions to improve the presentation of the conclusions. As an observational study, reviewers have also suggested using the STROBE checklist to improve the overall reporting of the study.

We look forward to receiving your revised manuscript.

Kind regards,

Emma Campbell, Ph.D

Staff Editor

PLOS ONE

Journal Requirements:

"Manyi Pan, Yuxin Zou, Gang Wei, Caoxu Zhang, Kai Zhang, Huaqi Guo and Weining Xiong declare that they have no known competing financial interests or personal relationships that could have appeared to influence the work reported in this paper."

Reviewers' comments:

Reviewer's Responses to Questions

**Comments to the Author**

1. Is the manuscript technically sound, and do the data support the conclusions?

Reviewer #1: Partly

Reviewer #2: Partly

2. Has the statistical analysis been performed appropriately and rigorously? 

Reviewer #1: I Don't Know

Reviewer #2: I Don't Know

3. Have the authors made all data underlying the findings in their manuscript fully available?

Reviewer #1: No

Reviewer #2: Yes

4. Is the manuscript presented in an intelligible fashion and written in standard English?

Reviewer #1: Yes

Reviewer #2: No

5. Review Comments to the Author

Reviewer #1: GENERAL COMMENTS

This is a study exploring potential associations between perfluoroalkyl substances (PFAS) exposure and the development of chronic obstructive pulmonary disease (COPD). Additionally, authors aimed to evaluate whether physical activity (PA) may protect individuals from developing COPD-related exposure to PFAS. They have used the 2013-2018 National Health and Nutrition Examination Survey (NHANES) data to perform analyses. The rationale for conducting the research has merits even if data and analysis are not robust for concluding on any cause-effect relationship. I miss, however, goodness of fit measures of the models they present.

SPECIFIC COMMENTS

Abstract

Line 28: This part of the sentence sound strange “…that have been studied to demonstrate their possible association with reduced lung function.”. Please improve. Also, begin the next sentence with a capital P (Physical activity).

Line 40: I find your conclusion overly cautious. I suggest something like: “PFAS exposure may increase the risk of developing COPD, but regular moderate-intensity physical activity can protect individuals from evolving to the disease. However, longitudinal studies are needed to support these preliminary findings.”

Introduction

Line 64: I suggest you use a more modern definition of PA, such as that by Piggin, J. (2020). What is physical activity? A holistic definition for teachers, researchers and policy makers. Frontiers in Sports and Active Living, 2, 72. https://www.frontiersin. org/articles/10.3389/fspor.2020.00072

Line 69: The rationale in the previous paragraphs direct the reader that PFAS and physical activity exposures can individually affect COPD in opposing directions, not just different directions.

Materials and Methods

Line 92: Please, add the specific link so that readers can access it easily.

Line 100: I suggest you delete “of sport”.

Line 106: I’m not aware that this article (reference #26) discusses the conversion of MET-min/week to 150min/week of moderate intensity activity. Can you tell in which page, please?

Lines 112–115: Can you provide the rationale and the supporting literature?

Line 121: “Therefore, ln-transformation was performed in the regression analysis to improve the normality”. It doesn’t matter the normality of the independent variable for regression analysis. What matters is the normality of the residuals or errors but in linear regression. More importantly, have you tested the assumptions for trusting the results of the logistic regression: linearity, independence of errors and multicollinearity? Please, report.

Line 126: Is “multinomial” missing before “logistic regression”? Or PA was an independent variable?

Results

Lines 133–134: I believe PA=0, PA=1 and PA=2 are probably the codes used as dependent variables in multinomial logistic regression models. I think they should be deleted. I think however that you should report more clearly in the statistical analysis section the dependent variable categories. You just state “various types of PA” on line 126.

Table 1. Please add the units of measurement in income and body mass index. Also, be consistent with the decimals. Sometimes, you don’t have any, present 1 or 2 decimals.

You have not prepared readers for sections 3.3. and 3.4, and Tables 2, 3 and 4 in the statistical analysis section. Please report there the methods and analyses used.

I’m not following you in section 3.4. Have you conducted separated analysis for each subgroup (just odds ratio) or have you conducted multivariate logistic regression?

Reviewer #2: This article presents interesting assumptions about COPD and physical activity. The work is relevant and with an adequate methodology, but some inaccurate elements need improvement. I have some major and minor comments listed below.

Major comments

#1 Revise English with a mother tongue reviewer.

#2 The hypothesis (lines 62 and 63) is poorly formulated. The relationship between PFAS and COPD was assessed, not the impact.

#3 The definition of physical activity is incomplete; I recommend an umbrella reference such as Caspersen et al. 1985. Additionally, reference 20 refers to a type of exercise and not physical activity. Several studies have proven the effectiveness of physical activity in COPD, that includes any bodily movement requiring energy expenditure.

#4 As an observational study, a guideline for reporting observational studies, such as the STOBE statement https://www.strobe-statement.org/, should be used, and authors should confirm through the checklist that all items are reported.

#5 The discussion is generally written with hasty conclusions and a lot of "certainty" instead of suggesting these relationships between PA, PFAS and COPD, as this study is observational.

Minor comments

#1 Revise acronyms throughout the manuscript. For example, the acronym for physical activity is defined in line 64 but is not used throughout the manuscript.

#2 Authors mention where they obtained the data, but URLs should be available (section 2.1).

#3 Typo in Figure 1: "analyse" instead of "analysis".

#4 In line 98, the "NHANES Physical Activity Questionnaire (PAQ)" is not described (type of question/ answer, scoring...).

#5 Line 112 - confound factors are mentioned, but not how they were obtained, whether through literature or what the reasoning was. Additionally, confirm that all variables are reported coherently in sections 2.4 and 2.5 (models 1 and 2) and the results table, and describe all variables in terms of their meaning (for example, the variable "Cycle" is not clear to what it refers to).

#6 Lines 124 - 125 - need to explain the rationale behind separating the confounding variables in the two models - statistical? Evidence-based?

#7 Table 1 - No indentation on the categories of physical activity.

#8 Lines 203-204 - Reformulate the sentence by removing "cessation".

6. PLOS authors have the option to publish the peer review history of their article (what does this mean?). If published, this will include your full peer review and any attached files.

Reviewer #1: **Yes: **Nuno Morais

Reviewer #2: No

---

## [Author Response · Author response to Decision Letter 0]

18 Jun 2024

Subject: Response to Review and Revisions for Manuscript ID [PONE-D-24-04889]

Dear Editor Emma Campbell and Reviewers,

 I trust this message finds you well. We sincerely appreciate the time and effort you have dedicated to reviewing our manuscript, 'Moderate-intensity physical activity reduces the role of serum PFAS on COPD: A cross-sectional analysis with NHANES data'. Your thoughtful insights have been invaluable, and we have carefully considered each point raised during the evaluation process. 

Reply to editor

Thank you for providing us with the feedback and additional requirements for the revision of our manuscript titled 'Moderate-intensity physical activity reduces the role of serum PFAS on COPD: A cross-sectional analysis with NHANES data' submitted to PLOS ONE. We appreciate the opportunity to address these points and ensure that our manuscript meets the journal's standards.

Response: We have carefully reviewed the PLOS ONE style templates provided and have made the necessary adjustments to ensure that our manuscript complies with the journal's style requirements. The main body and title/authors/affiliations sections have been formatted accordingly.

Response: We apologize for the discrepancy in the grant information provided in the 'Funding Information' and 'Financial Disclosure' sections. We have revised the 'Funding Information' section to accurately reflect the grant numbers for the awards received for our study. This work was financially supported by the National Natural Science Foundation of China (No. 82090015). We have ensured that these revisions are reflected in the submission system.

"Manyi Pan, Yuxin Zou, Gang Wei, Caoxu Zhang, Kai Zhang, Huaqi Guo and Weining Xiong declare that they have no known competing financial interests or personal relationships that could have appeared to influence the work reported in this paper." Please complete your Competing Interests on the online submission form to state any Competing Interests. If you have no competing interests, please state "The authors have declared that no competing interests exist.", as detailed online in our guide for authors at http://journals.plos.org/plosone/s/submit-now

Response: We apologize for any oversight in our previous submission. Thank you for pointing this out. We have updated the Competing Interests section on the online submission form to state that "The authors have declared that no competing interests exist," in accordance with the journal's guidelines. This information has also been included in our cover letter for your reference.

Response: Thank you for bringing this to our attention. We apologize for any oversight. Upon review, we have determined that the data referenced by the phrase "data not shown" do not pertain to the core parts of our research. Specifically, the data indicated that serum PFAS concentrations were not associated with age or the difference between current age and age of COPD diagnosis. This was included as evidence against reverse causation but is not central to our main findings on PFAS and COPD. Therefore, to comply with your guidelines, we have removed the relevant content from the manuscript. 

We appreciate your attention to this matter and look forward to your continued feedback.

Reply to Review 1

 Thank you for your insightful review of our manuscript. We appreciate your recognition of the rationale behind our study exploring the potential links between PFAS exposure and COPD development, as well as the role of physical activity. We acknowledge your suggestion regarding the inclusion of goodness of fit measures for our models, and we have addressed this in our revision. Your feedback is invaluable, and we are committed to enhancing the quality of our research based on your suggestions. 

1. Line 28: This part of the sentence sound strange “…that have been studied to demonstrate their possible association with reduced lung function.”. Please improve. Also, begin the next sentence with a capital P (Physical activity).

Line 40: I find your conclusion overly cautious. I suggest something like: “PFAS exposure may increase the risk of developing COPD, but regular moderate-intensity physical activity can protect individuals from evolving to the disease. However, longitudinal studies are needed to support these preliminary findings.”

 Response: Thank you for your valuable feedback. I have revised the sentence on line 28 to enhance clarity. The updated sentence reads: "Perfluoroalkyl substances (PFAS) are synthetic chemicals known for their high stability and durability. Research has examined their potential link to decreased lung function." Additionally, "Physical activity" has been capitalized at the beginning of the next sentence. The detailed changes can be found in bold in the revised manuscript with track changes, specifically on line 22 to 24.

I appreciate your suggestion to make the conclusion more assertive. Here is the revised version: " PFAS exposure may increase the risk of developing COPD, but regular moderate-intensity physical activity can protect individuals from evolving to the disease. However, longitudinal studies are needed to support these preliminary findings.." Detailed information can be found in Bold in Revised Manuscript with Track, specifically line 36 to 37.

2. Line 64: I suggest you use a more modern definition of PA, such as that by Piggin, J. (2020). What is physical activity? A holistic definition for teachers, researchers and policy makers. Frontiers in Sports and Active Living, 2, 72. https://www.frontiersin. org/articles/10.3389/fspor.2020.00072IF: 2.7

Response：Thank you for reviewing our manuscript and providing valuable feedback. We highly appreciate your suggestion and would like to offer further clarification regarding our choice of definition for physical activity (PA).

You suggested using the definition: “Physical activity involves people moving, acting and performing within culturally specific spaces and contexts, and influenced by a unique array of interests, emotions, ideas, instructions, and relationships.” This is indeed a valuable definition, particularly for exploring the cultural and social contexts of physical activity. However, we have chosen to use the definition: “Physical activity (PA) is an activity designed to promote health, strengthen fitness, and enrich life through a variety of sports according to physical needs.” This definition aligns more closely with the specific context and methodology of our research, which is based on data from the NHANES database.

Our study primarily relies on the NHANES database and uses MET as the unit of measurement to quantify physical activity levels. Therefore, the definition provided in "Increasing physical activity in the community setting" is more appropriate for our study. This definition has also been utilized in key references for our research, including:

Physical activity reduces the role of blood cadmium on depression: A cross-sectional analysis with NHANES data [1]

The Moderating Effect of Physical Activity on the Relationship Between Neutrophils and Depression: A Cross-sectional Analysis Based on the NHANES Database [2]

These references employ the same definition and use MET as the unit of measurement, ensuring consistency and reliability in our research methodology. We also noted that the papers citing your suggested definition do not use MET as a unit nor do they use the NHANES database. This distinction further supports our choice of definition to maintain consistency with our data source and analytical methods.

Additionally, we have expanded the definition of PA in our manuscript to: "Physical activity (PA) includes any bodily movement produced by skeletal muscles that requires energy expenditure. It encompasses various activities across four domains: occupational (work-related), domestic (household chores), transportation (walking or cycling to get from one place to another), and leisure time (recreational activities)." Detailed information can be found in bold in the revised manuscript with track changes, specifically on lines 63 to 65.

We hope this explanation clarifies our rationale for selecting this particular definition. If you have any further questions or suggestions, we would be happy to discuss them.We hope this explanation clarifies our rationale for selecting this particular definition. If you have any further questions or suggestions, we would be happy to discuss them.

3. Line 69: The rationale in the previous paragraphs direct the reader that PFAS and physical activity exposures can individually affect COPD in opposing directions, not just different directions.

Response: Thank you for your observation. The rationale presented in the preceding paragraphs suggests that exposures to PFAS and physical activity may individually influence COPD in opposing directions, rather than simply different directions. Specifically, we have revised the sentence to: "Since PFAS and PA can individually affect COPD in opposite directions" in the paper. Detailed information can be found in Bold in Revised Manuscript with Track, specifically line 69.

4. Line 92: Please, add the specific link so that readers can access it easily.

Response: Thank you for your insightful review. In response to your request for additional information, we have provided the specific link for easy access: https://wwwn.cdc.gov/Nchs/Nhanes/2017-2018/PFAS_J.htm. Detailed information can be found in Bold in Revised Manuscript with Track, specifically line 93.

5.Line 100: I suggest you delete “of sport”.

Response: For line 100, I appreciate your suggestion to delete "of sport." We have made the suggested change and delete "of sport" from the definition. Detailed information can be found in Bold in Revised Manuscript with Track, specifically line 103 to 104.

6.Line 106: I’m not aware that this article (reference #26) discusses the conversion of MET-min/week to 150min/week of moderate intensity activity. Can you tell in which page, please?

Response：Thank you for your insightful feedback. Upon reviewing reference #26, I realize it does not specifically discuss the conversion of MET-min/week to 150 minutes/week of moderate-intensity activity. I apologize for this oversight. We doublecheck all of the citations to make sure they are properly cited. To address this issue, I have amended the manuscript by adding appropriate references that clearly explain this conversion. Specifically, the conversion is based on established guidelines, such as those from the World Health Organization (WHO) and other authoritative sources. 

On the first page of World Health Organization 2020 guidelines on physical activity and sedentary behavior, it states [3], "All adults should undertake 150-300 minutes of moderate-intensity physical activity per week." Additionally, Reference #25 explains, "Each intensity was assigned a corresponding metabolic equivalent (MET, 1 MET = 1 kilocalorie per hour per kilogram of bodyweight): 3.3 MET for walking, 4.0 MET for moderate PA, and 8.0 MET for vigorous PA. We then quantified PA of each participant by calculating minutes of MET each week (MET-min/week) based on the reported intensity, duration, and frequency of PA in 1 week." [4]. By dividing 600 MET-minutes per week by 4.0, we obtain 150 minutes per week, which is also mentioned on the third page of Reference #27 [5]. Detailed information can be found in Bold in Revised Manuscript with Track, specifically line 109 to 110.

7. Lines 112–115: Can you provide the rationale and the supporting literature?

Response: Thank you for your question. We considered adjusting for variables associated with the diagnosis of COPD or those that confound the relationship between serum PFAS concentrations and the diagnosis of COPD. These variables include age, race, gender, BMI, physical activity, health insurance coverage, NHANES cycle, alcohol consumption, marital status, and household income.

The rationale for including these variables is based on their potential to influence COPD outcomes and their established relevance in existing literature. For example, age, race, gender, BMI, and physical activity are well-documented factors that can affect COPD diagnosis and progression. Health insurance coverage and socioeconomic factors like household income and marital status also play significant roles in health outcomes and access to care.

Supporting literature includes references such as Cai et al. (2014) [6], which discusses the impact of socioeconomic status and air pollution on respiratory health, including COPD, and Garcia-Aymerich et al. (2007) [7], which addresses the influence of physical activity on COPD risk and lung function decline. Detailed information can be found in Bold in Revised Manuscript with Track, specifically line 117 to 119.

8.Line 121: “Therefore, ln-transformation was performed in the regression analysis to improve the normality”. It doesn’t matter the normality of the independent variable for regression analysis. What matters is the normality of the residuals or errors but in linear regression. More importantly, have you tested the assumptions for trusting the results of the logistic regression: linearity, independence of errors and multicollinearity? Please, report.

Response: Thank you for your insightful comments and the opportunity to improve our manuscript. We appreciate your feedback regarding the assumptions underlying the logistic regression analysis. 

The normality of the independent variables is not a concern. What matters for regression analysis is the normality of the residuals or errors, especially in linear regression. For logistic regression, it is more important to test specific assumptions. Below, we address your specific concerns:

Residual Independence

In logistic regression, the errors are assumed to follow a binomial distribution, and the distribution of the independent variables is not the main concern. Therefore, the normality of residuals is not required. However, the independence of errors is crucial, as it affects the bias and validity of the model estimates. We used the Autocorrelation Function (ACF) plot to check for error independence.

Linearity

For continuous variables, we used diagnostic tools like the Component + Residual (CR) plot to examine the linear relationship between each predictor and the log-odds. For binary or categorical predictors, this "linearity" test is not applicable, as categorical variables do not involve a linear relationship across a numerical range. The logistic regression model is designed to assess the difference in log-odds effects across different categories.

Multicollinearity

We continued to use the Variance Inflation Factor (VIF) to check for collinearity. A VIF v

---

## [Decision Letter · Decision Letter 1]

18 Jul 2024

Moderate-intensity physical activity reduces the role of serum PFAS on COPD: A cross-sectional analysis with NHANES data

PONE-D-24-04889R1

Dear Dr. Guo,

We’re pleased to inform you that your manuscript has been judged scientifically suitable for publication and will be formally accepted for publication once it meets all outstanding technical requirements.

Kind regards,

Vanessa Carels, PhD

Staff Editor

PLOS ONE

Additional Editor Comments (optional):

Please attend to Reviewer #2's minor comment regarding some missing acronyms throughout the manuscript prior to final submission, thank you!

Reviewers' comments:

Reviewer's Responses to Questions

**Comments to the Author**

1. If the authors have adequately addressed your comments raised in a previous round of review and you feel that this manuscript is now acceptable for publication, you may indicate that here to bypass the “Comments to the Author” section, enter your conflict of interest statement in the “Confidential to Editor” section, and submit your "Accept" recommendation.

Reviewer #1: All comments have been addressed

Reviewer #2: All comments have been addressed

2. Is the manuscript technically sound, and do the data support the conclusions?

Reviewer #1: Yes

Reviewer #2: (No Response)

3. Has the statistical analysis been performed appropriately and rigorously? 

Reviewer #1: Yes

Reviewer #2: (No Response)

4. Have the authors made all data underlying the findings in their manuscript fully available?

Reviewer #1: Yes

Reviewer #2: (No Response)

5. Is the manuscript presented in an intelligible fashion and written in standard English?

Reviewer #1: Yes

Reviewer #2: (No Response)

6. Review Comments to the Author

Reviewer #1: (No Response)

Reviewer #2: The comments have been addressed and the paper has improved significantly, being acceptable for publication. I would just leave a small suggestion to repeat the search for the words "physical activity" throughout the article, as there are still some acronyms missing. Congratulations on your work and dedication!

7. PLOS authors have the option to publish the peer review history of their article (what does this mean?). If published, this will include your full peer review and any attached files.

Reviewer #1: **Yes: **Nuno Morais

Reviewer #2: No

---

## [Editor Report · Acceptance letter]

26 Jul 2024

PONE-D-24-04889R1 

PLOS ONE

Dear Dr. Guo, 

I'm pleased to inform you that your manuscript has been deemed suitable for publication in PLOS ONE. Congratulations! Your manuscript is now being handed over to our production team.

Kind regards, 

on behalf of

Dr. Avanti Dey 

Staff Editor

PLOS ONE